

# Does size matter? An analysis of the niche width and vulnerability to climate change of fourteen species of the genus *Crotalus* from North America

Jorge Luis Becerra-López[1], Raciel Cruz-Elizalde[2], Aurelio Ramírez-Bautista[3], Itzel Magno-Benítez[3], Claudia Ballesteros-Barrera[4], Javier Alvarado-Díaz[5], Robert W. Bryson Jr[6], Uriel Hernández-Salinas[7], César A. Díaz-Marín[3], Christian Berriozabal-Islas[8,9], Karen Fraire-Galindo[1], Juan Tello-Ruiz[1], Alexander Czaja[1] and María Guadalupe Torres-Delgado[1]

[1] Laboratorio de Cambio Climático y Conservación de Recursos Naturales, Facultad de Ciencias Biológicas, Universidad Juárez del Estado de Durango, Gómez Palacio, Durango, Mexico
[2] Laboratorio de Zoología, Facultad de Ciencias Naturales, Universidad Autónoma de Querétaro, Querétaro, Querétaro, Mexico
[3] Laboratorio de Ecología de Poblaciones, Centro de Investigaciones Biológicas, Instituto de Ciencias Básicas e Ingeniería, Universidad Autónoma del Estado de Hidalgo, Mineral de La Reforma, Hidalgo, Mexico
[4] Unidad Iztapalapa, División de Ciencias Biológicas y de la Salud, Universidad Autónoma Metropolitana, Ciudad de México, Ciudad de México, Mexico
[5] Instituto de Investigaciones sobre los Recursos Naturales, Universidad Michoacana de San Nicolás de Hidalgo, Morelia, Michoacan, Mexico
[6] Sierra Madre Research Institute, San Antonio, TX, United States of America
[7] CIIDIR Unidad Durango, Instituto Politécnico Nacional, Durango, Durango, Mexico
[8] Universidad Tecnológica de la Zona Metropolitana del Valle de México, Miguel Hidalgo y Costilla No. 5, Fraccionamiento Los Héroes, Tizayuca, Hidalgo, México
[9] Universidad de Quintana Roo, Departamento de Administración turística, Playa del Carmen, Cancún, Quintana Roo, México

Corresponding author
Aurelio Ramírez-Bautista,
ramibautistaa@gmail.com

## ABSTRACT

The niche comprises the set of abiotic and biotic environmental conditions in which a species can live. Consequently, those species that present broader niches are expected to be more tolerant to changes in climatic variations than those species that present reduced niches. In this study, we estimate the amplitude of the climatic niche of fourteen species of rattlesnakes of the genus *Crotalus* to evaluate whether those species that present broader niches are less susceptible to the loss of climatically suitable zones due to the projected climate change for the time period 2021–2040. Our results suggest that for the species under study, the breadth of the niche is not a factor that determines their vulnerability to climatic variations. However, 71.4% of the species will experience increasingly inadequate habitat conditions, mainly due to the increase in temperature and the contribution that this variable has in the creation of climatically suitable zones for most of these species.

## INTRODUCTION

Global climate change is one of the main factors that impact biodiversity and the distribution of species (*Barnosky et al., 2011*). Each species has a tolerance to various environmental factors, and when this tolerance is exceeded, the species cannot optimally carry out their life cycle (*Peters, 1990*; *Walther et al., 2002*; *Hardy, 2003*; *Dawson & Spannagle, 2009*). When this occurs, the distribution and abundance of the species is altered (*Hughes, 2000*; *Peterson et al., 2005*; *Root et al., 2005*; *Parmesan, 2006*), and in some cases, it can result in the direct disappearance of some species and populations (*Walther et al., 2002*; *Thomas et al., 2004*). This in turn creates conditions that could modify the structure in the composition of species in the ecosystem and, consequently, disturb the ecological balance of a landscape (*Gray, 2005*; *Walther, Beißner & Burga, 2005*).

Niche modeling provides a predictive measure about how the climatic suitability of a species may change under different climate change scenarios (*Morin & Lechowicz, 2008*; *Thuiller, Lavorel & Araújo, 2005b*; *Lawler et al., 2009*). Currently, most niche models have been developed from a correlative approach, particularly when more than one species is involved (*Hijmans & Graham, 2006*). In this approach, the environmental variables that characterize the places where a species occurs (or is absent) are used to develop correlative models that analyze the effect of climate change on the species' climatic suitability (*Wiens et al., 2009*).

Rattlesnakes of the genus *Crotalus* are widely distributed across the New World from southern Canada to Argentina (*Campbell & Lamar, 2004*). There are currently 53 recognized species, with the greatest number found in Mexico (*Sánchez et al., 2020*). Various authors point out that temperature and precipitation are important factors in the ecology of the species of this genus (*Paredes-García, Ramírez-Bautista & Martínez-Morales, 2011*; *Sunny et al., 2019*; *Yañez-Arenas et al., 2020*). As such, *Crotalus* represent a good model to predict the response of snake species to climate change. However, there are few studies that evaluate the effects that these environmental variations will have on the future distributions of species of this genus (*Greene & Campbell, 1993*; *Gibbons et al., 2000*). In this regard, and under the criterion that the niche comprises a set of environmental conditions in which a species may exist (*Gaston, Blackburn & Lawton, 1997*), it has been suggested that those species with broader niches could be less vulnerable to abrupt environmental variation under anthropogenic climate change. By contrast, those species with narrow niches would be particularly threatened by climatic disturbances (*Brown, 1984*; *Johnson, 1998*; *Boyles & Storm, 2007*; *Botts, Erasmus & Alexander, 2013*; *Ozinga et al., 2013*).

From this perspective, the question arises: can the breadth of niche, by itself, be considered as a determining factor that helps to predict the vulnerability of *Crotalus* species to climate change? Few studies have provided sufficient evidence to answer this question and thus the effects that climate change will have on each of the species of this genus, remain unknown (*Greene & Campbell, 1993*; *Gibbons et al., 2000*). The present study aims to analyze whether there is a relationship between niche width and vulnerability to climate change, projected for the period 2021–2040, in a sample of fourteen species of the genus

*Crotalus* distributed in North America. This information is of great relevance for the establishment and development of conservation strategies for species of the genus *Crotalus*.

## MATERIAL AND METHODS

### Presence data

We obtained geographical data of occurrences of 14 species of *Crotalus*, including *C. atrox*, *C. basiliscus*, *C. cerastes*, *C. enyo*, *C. intermedius*, *C. lepidus*, *C. molossus*, *C. pricei*, *C. ravus*, *C. ruber*, *C. scutulatus*, *C. tigris*, *C. viridis*, and *C. willardi* (following the taxonomy of *Campbell & Lamar, 2004*). We obtained geographical data from published scientific information (scientific articles, scientific notes, books), scientific collections from Mexico and other countries (Table S1), information generated by the National Commission of Protected Natural Areas (CONANP), as well as from the database of the Global Biodiversity Information Facility (GBIF; http://www.data.gbif.org). We selected these 14 species of the genus *Crotalus* because, after the geographic data purification process, they were the species that had the most complete base of geographic records with the best distributed geographic records in the known range of these species, reflecting with greater precision the total range of the species under study (*Campbell & Lamar, 2004*). As has been previously demonstrated, the precision of geographic records is of great relevance in the performance of species distribution models (*Hefley et al., 2014*; *Fei & Yu, 2015*; *Velásquez-Tibatá, Graham & Munch, 2015*). Data 'cleanliness' is particularly important for data coming from species distribution data warehouses such as GBIF (*Hijmans & Elith, 2013*). Using the "dismo" library (*Hijmans et al., 2017*) in the statistical software R (version 3.1.3, *R Core Team, 2015*), we checked the geographic projections of each record and eliminated duplicate records. We further cross-checked coordinates through visual inspection (*Hijmans et al., 1999*) and assessed sampling bias by subsampling the geographic records (*Hijmans & Spooner, 2001*; *Phillips et al., 2009*). Records with unreliable coordinates (according to the known distribution of the species; *Campbell & Lamar, 2004*) were removed from the database. In total, we generated a data set with 4,813 presence points (*C. atrox* = 1,241, *C. basiliscus* = 125, *C. cerastes* = 676, *C. enyo* = 135, *C. intermedius* = 41, *C. lepidus* = 239, *C. molossus* = 516, *C. pricei* = 76, *C. ravus* = 52, *C. ruber* = 568, *C. scutulatus* = 610, *C. tigris* = 72, *C. viridis* = 429, and *C. willardi* = 33; Fig. 1).

### Climatic variables

Current weather data for North America was obtained with a resolution of 2.5 min (~5 km) from the WorldClim database (version 2). This is an online database with 19 bioclimatic variables derived from monthly averages (1970–2000) of temperature and precipitation (*Fick & Hijmans, 2017*). We selected a subset of these variables on the basis of ecological theory and subsequently reduced, when necessary, through statistical analysis (*Austin, 2007*). In the preselection of the variables related to temperature, we considered those proposed by *Rodder & Lotters (2009)*, who suggested that this set of variables were of great ecological relevance, particularly for those taxa limited by thermoregulation, such as squamates. The variables related to precipitation included descriptors that have been mentioned as key factors for the species of the genus *Crotalus*, which may become more

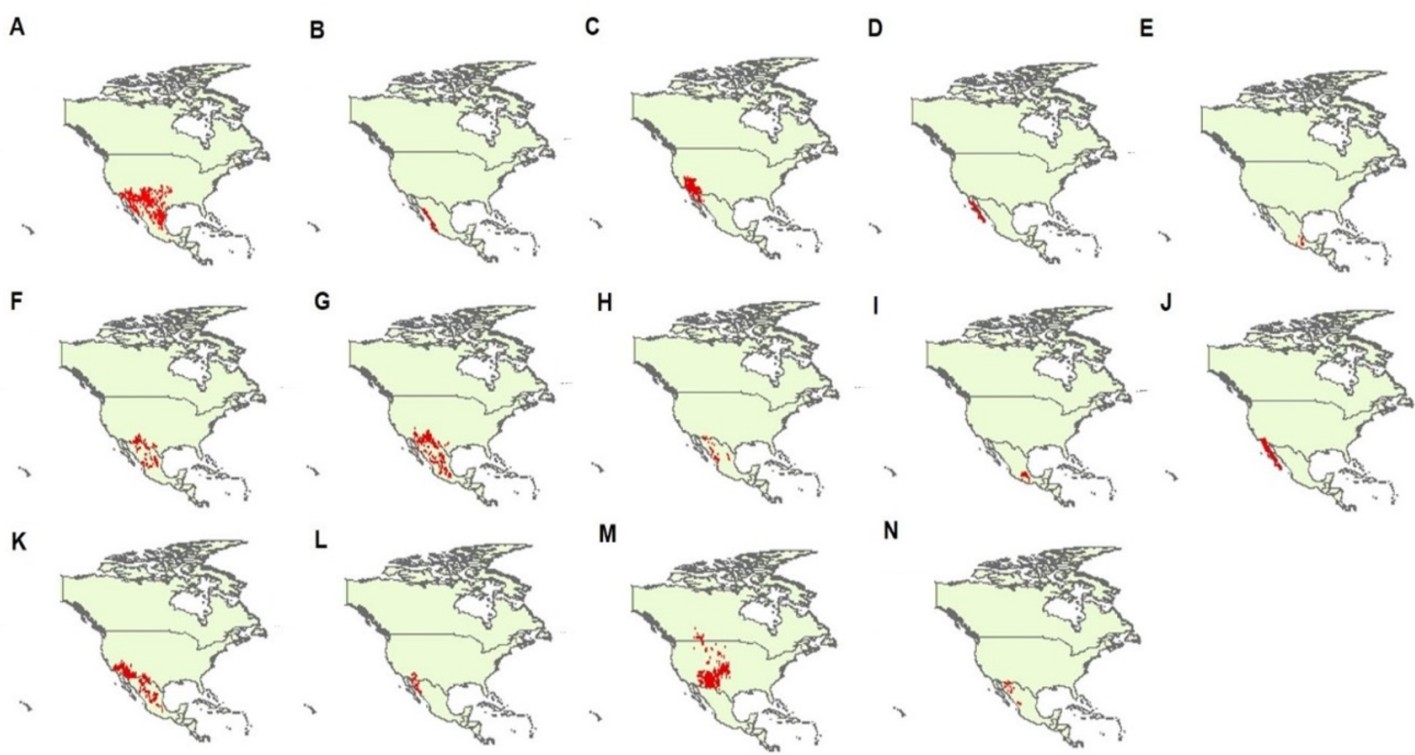

**Figure 1** **Geographic records of 14 *Crotalus* species used in this study.** Species are (A) *C. atrox*, (B) *C. basiliscus*, (C) *C. cerastes*, (D) *C. enyo*, (E) *C. intermedius*, (F) *C. lepidus*, (G) *C. molossus*, (H) *C. pricei*, (I) *C. ravus*, (J) *C. ruber*, (K) *C. scutulatus*, (L) *C. tigris*, (M) *C. viridis*, and (N) *C. willardi*. Taxonomy follows *Campbell & Lamar (2004)*. Red dots denoted each geographic record for each species analyzed in this study.

relevant when thermal conditions are not optimal, for example in periods of time with extreme temperatures (*Glaudas, 2009*; *Phadnis et al., 2019*). Subsequently, to eliminate variables that provide similar information, we developed a Pearson correlation matrix ($r < 0.7$) to reduce the collinearity error.

After this process, the retained variables were Annual Mean Temperature (bio1), Mean Diurnal Range (bio2), Mean Temperature of Wettest Quarter (bio8), Annual Precipitation (bio12), Precipitation of Wettest Month (bio13), Precipitation of Driest Month (bio14), Precipitation Seasonality (bio15), Precipitation of Warmest Quarter (bio18) and Precipitation of Coldest Quarter (bio19). In general, the bivariate correlation analysis was carried out by providing information on the 19 climatic variables to the presence records of the species under study. In our case, the climatic information was provided to 10,000 randomly distributed geographic points in the distribution area of the species under study to avoid discarding areas with relevant climate information (non-repetitive) (*Becerra-López et al., 2016*).

## Climate profile and niche range

With the selected variables of the current climate, a Principal Component Analysis (PCA) was carried out in the software R using the "ecospat" library (*Broennimann et al., 2014*) to identify the climatic profile within the distribution area of the species under study.

We also evaluated the climate profile for the climate change models BCC-CSM2-MR, CNRM-CM6-1, and IPSL-CM6A-LR, considering the shared socio-economic pathway 5 8.5 W/m$^2$ (SSP5 8.5) proposed for the period 2021–2040. These climate models were randomly selected from a total of eight models.

For each selected variable, we then performed an Analysis of Variance and Tukey's post hoc tests to evaluate if there were statistical differences between the current climate data and the climate change scenarios. Subsequently, in the software R, the distribution of the species under study in the climatic space (niche range) were identified through a PCA using the nine current climate variables used in this study, following the methodology proposed by *Becerra-López et al. (2020)*. This representation of the records of the species in a climatic context is based on the Hutchinson duality that indicates that there are two spaces, the geographic one and the multidimensional abstract space, denoted by climatic variables that establish the conditions in which a species can simply exist (*Colwell & Rangel, 2009*).

For the selection of SSP5 8.5 W/m$^2$, we took into account that the narrative of this route considers a socioeconomic development driven by fossil fuels, which implies a scenario with increasing $CO_2$ emissions (*Riahi et al., 2016*; *Kriegler et al., 2017*). Considering that fossil fuels meet current energy demand, and it is estimated that they will supply at least 80% of the energy demand required in 2040 (*Beltrán-Telles et al., 2017*), we decided to use only SSP5 8.5 W/m$^2$ to model the availability of suitable climatic environments for the presence of the species under study. Likewise, we considered that SSP5 8.5 W/m$^2$ is the climate scenario that provides the best test of our hypothesis as this will result in the largest difference between current and future environmental conditions.

## Vulnerability of climatic suitability in the face of environmental variations

The Maximum Entropy (MaxEnt) approach was used to model the climatic suitability of the 14 species of *Crotalus*. MaxEnt uses the principle of maximum entropy on presence-only data to estimate a set of functions that relate environmental variables and climatic suitability to approximate the species' niche and potential geographic distribution (*Phillips et al., 2017*). Therefore, the species distribution model considered in this study represents a correlative species distribution model (*Phillips, Anderson & Schapire, 2006*), subject to the challenge of balancing goodness of fit with model complexity, as models that are inappropriately complex or simple show reduced ability to infer habitat quality, reduced ability to infer relative importance of the variables in the restriction of the distribution of the species, and a reduced transferability to other time periods (*Warren & Seifert, 2011*). In our case study, using the "ENMeval" library (*Muscarella et al., 2014*) in the software R, the calibration of the model for each species considered the choice of (a) accessible area (background or M area), (b) the type of variables that MaxEnt constructs (features), (c) regularization multiplier, and (d) the type of model output (raw, cumulative, logistic), as these considerations affect the inferences to be made (*Fourcade et al., 2014*).

Using the MaxEnt software, the information obtained from the calibrated models was projected within the known distribution area of the species under study. We used the layers of the current climate and those of the future climate mentioned above. All climatic
layers were obtained from the WorldClim database v2.1 (https://www.worldclim.org/). The models were generated with a climatic suitability gradient from 0 (low suitability) to 1 (high suitability), which were then converted to binary models (presence/absence). For each species, the threshold Maximum training sensitivity plus specificity (MaxSS) provided by MaxEnt in each model was chosen. The threshold MaxSS has been reported to show good performance for models that work only with presence data (*Liu, White & Newell, 2013*). The importance of each bioclimatic variable in the observed distribution of the species under study was evaluated according to the relative importance of each variable, which was obtained by adding the percentage of contribution (PC) and the importance of permutation (IP), evaluated by MaxEnt, and the result was divided by two $[\frac{average\ contribution(PC+IP)}{2}]$ (*Anadón et al., 2015*).

As a last step, the climatic suitability of the realized niche of each species was measured under current and future climate conditions. The vulnerability of the climatic suitability of each species to climate change was also identified, using the following change rate analysis: $\%\ of\ change = \left[\frac{(S1-S0)}{S0}\right]*100$, where $S0$ is the total surface of the study area, according to the base scenario, and $S1$ is the total surface occupied in the study area under change conditions.

## RESULTS

### Climate profile

The PCA suggested that, for our study area, the climate profile could be explained by considering the first two components. In all cases between components one and two, they explained at least 95% of the variation in the data. Under current weather conditions, for example, component one explained 96.2% of this variation, while component two only explained 2.8%. Considering the climate change scenarios, the scenario that presented the value with the lowest percentage in the sum of the two components was the BCC-CSM2-MR scenario with a value of 95.1%. The highest value was presented in the CNRM-CM6-1, and IPSL-CM6A-LR scenarios with 96%.

Regarding the contribution of the variables for each component, for both the current climate conditions and the climate change scenarios, the variable Annual Precipitation was the one that presented the greatest contribution in component one. For component two, considering the current climate conditions and the climate change scenarios, the variables Precipitation of Warmest Quarter and Precipitation of Coldest Quarter were the ones that presented the greatest contribution (Table 1); however, the Analysis of Variance and Tukey post hoc tests suggested that only the climatic variables Annual Mean Temperature and Mean Temperature of Wettest Quarter presented significant statistical differences in their means with respect to the three climate change scenarios used in this study. While the variable Mean Diurnal Range only presented significant differences in its means when contrasted with the climatic information proposed for the scenarios BCC-CSM2-MR and CNRM-CM6-1, the rest of the variables did not present significant differences (Table 2).

Regarding the size of the niches, our results showed that these amplitudes varied among components. For example, *C. ravus* presented the greatest niche width considering the

**Table 1 Contribution values of the climate variables for each component of the three climate change scenarios projected for the period 2021–2040 (Socio-economic Pathways (SSPs): 585).** The magnitude is used to choose the variables that best explained most of the variation, which is $\geq 0.50$. Climate change scenarios correspond to BCC-CSM2-MR, CNRM-CM6-1, IPSL-CM6A-LR. Variables are bio1 = Annual Mean Temperature, bio2 = Mean Diurnal Range, bio8 = Mean Temperature of Wettest Quarter, bio12 = Annual Precipitation, bio13 = Precipitation of Wettest Month, bio14 = Precipitation of Driest Month, bio15 = Precipitation Seasonality, bio18 = Precipitation of Warmest Quarter, bio19 = Precipitation of Coldest Quarter.

| Variables | Current weather | | BCC-CSM2-MR | | CNRM-CM6-1 | | IPSL-CM6A-LR | |
|---|---|---|---|---|---|---|---|---|
| | PC 1 | PC 2 | PC 1 | PC 2 | PC 1 | PC 2 | PC 1 | PC 2 |
| bio1 | 0.013 | −0.02 | 0.006 | −0.013 | 0.005 | −0.014 | 0.006 | −0.015 |
| bio2 | 0.0004 | 0 | −0.002 | −0.005 | −0.002 | −0.004 | −0.002 | −0.005 |
| bio8 | 0.004 | −0.04 | 0.004 | −0.046 | 0.003 | −0.049 | 0.004 | −0.047 |
| bio12 | 0.94 | −0.03 | 0.93 | 0.22 | 0.93 | 0.206 | 0.941 | 0.197 |
| bio13 | 0.12 | −0.07 | 0.17 | −0.24 | 0.177 | −0.237 | 0.169 | −0.223 |
| bio14 | 0.038 | 0.015 | 0.02 | 0.09 | 0.024 | 0.097 | 0.022 | 0.095 |
| bio15 | −0.017 | −0.03 | 0.01 | −0.22 | 0.013 | −0.215 | 0.01 | −0.214 |
| bio18 | 0.18 | −0.69 | 0.3 | −0.72 | 0.275 | −0.74 | 0.272 | −0.741 |
| bio19 | 0.24 | 0.70 | 0.09 | 0.55 | 0.096 | 0.542 | 0.097 | 0.55 |

**Table 2 The significance values of the Analysis of Variance (ANOVA) for each climatic variable identifying if at least one of the three climate scenarios projected for the 2021–2040 period differs from the current climate.** Likewise, the significance value of the Tukey post hoc test is shown, identifying which climatic scenario is the one that presents these variations. Climate change scenarios are BCC-CSM2-MR (A), CNRM-CM6-1, (B) and IPSL-CM6A-LR (C). Variables are bio1 = Annual Mean Temperature, bio2 = Mean Diurnal Range, bio8 = Mean Temperature of Wettest Quarter, bio12 = Annual Precipitation, bio13 = Precipitation of Wettest Month, bio14 = Precipitation of Driest Month, bio15 = Precipitation Seasonality, bio18 = Precipitation of Warmest Quarter, bio19 = Precipitation of Coldest Quarter.

| Variables | ANOVA | Tukey *post hoc* | | |
|---|---|---|---|---|
| | Current weather *vs.* future | (A) | (B) | (C) |
| bio1 | $F = 17.234$, g.l. = 3, 3704; $P < 0.001$ | 0 | 0 | 0 |
| bio2 | $F = 11.024$, g.l. = 3, 3704; $P < 0.001$ | 0 | 0 | 0.706 |
| bio8 | $F = 9.164$ , g.l. = 3, 3704; $P < 0.001$ | 0.009 | 0 | 0 |
| bio12 | $F = 0.646$, g.l. = 3, 3704; $P = 0.585$ | 0.659 | 1 | 0.929 |
| bio13 | $F = 2.246$, g.l. = 3, 3704; $P = 0.081$ | 0.071 | 0.935 | 0.94 |
| bio14 | $F = 0.056$, g.l. = 3, 3704; $P < 0.921$ | 0.995 | 0.993 | 0.978 |
| bio15 | $F = 1.847$, g.l. = 3, 3704; $P = 0.133$ | 0.146 | 0.997 | 0.931 |
| bio18 | $F = 2.205$, g.l. = 3, 3704; $P = 0.085$ | 0.527 | 0.619 | 0.999 |
| bio19 | $F = 0.065$, g.l. = 3, 3704; $P = 0.978$ | 0.984 | 0.98 | 0.997 |

principal component one, with a range from −67.96149 to 1318.77525. In component two, this species occupied the third position in descending order, with a range from −176.6954 to 109.6385. *Crotalus basiliscus*, on the other hand, was in the second position in niche width in component one, with a range from 30.20758 to 1216.04195; in component two, this species was in the first position with a range from −616.705 to −101.3538. For species that presented the lowest niche amplitudes, *C. cerastes* showed in component one a range

**Table 3** **The niche amplitude ranges of the *Crotalus* species under study for each component.** Amplitude level is assigned with the numbering from 1 to 14, considering the value 1 as the greatest amplitude and the value 14 as the least amplitude.

| Amplitude level | Species | Principal component 1 | | Species | Principal component 2 | |
|---|---|---|---|---|---|---|
| 1 | *Crotalus ravus* | −67.96149 | 1318.77525 | *C. basiliscus* | −616.705 | −101.3538 |
| 2 | *C. basiliscus* | 30.20758 | 1216.04195 | *C. ruber* | −163.621 | 178.7878 |
| 3 | *C. lepidus* | −287.7632 | 861.3883 | *C. ravus* | −176.6954 | 109.6385 |
| 4 | *C. atrox* | −465.0134 | 526.5947 | *C. enyo* | −178.12807 | 82.67381 |
| 5 | *C. pricei* | −109.8672 | 848.9207 | *C. lepidus* | −250.781818 | 9.362773 |
| 6 | *C. intermedius* | −24.45662 | 857.09356 | *C. scutulatus* | −108.6258 | 132.9914 |
| 7 | *C. molossus* | −292.5419 | 492.205 | *C. molossus* | −164.95542 | 49.20508 |
| 8 | *C. willardi* | −90.90045 | 565.84212 | *C. tigris* | −146.12779 | 60.16305 |
| 9 | *C. scutulatus* | −461.1022 | 91.95143 | *C. atrox* | −110.62657 | 84.72098 |
| 10 | *C. tigris* | −340.6544 | 164.7327 | *C. pricei* | −201.97193 | −15.33427 |
| 11 | *C. viridis* | −316.2345 | 130.2748 | *C. willardi* | −186.64149 | −21.56087 |
| 12 | *C. enyo* | −476.14736 | −34.06174 | *C. cerastes* | −22.74158 | 118.81858 |
| 13 | *C. ruber* | −464.42653 | −43.88676 | *C. intermedius* | −74.70301 | 52.86688 |
| 14 | *C. cerastes* | −490.6939 | −197.8326 | *C. viridis* | −60.4701 | 55.13558 |

from −490.6939 to −197.8326, placing it in position 14. In component two, this species was in the position number 12 with a niche width range from −22.74158 to 118.81858 (see Table 3).

## Vulnerability of climatic suitability in the face of environmental variations

The models obtained for the species under study showed an area under the curve ranging from 0.80 to 0.95, indicating low levels of commission (predicts the presence of the species where it does not exist, false positive) and omission (predicts the non-presence of the species where it really exists, false negative) (Table 4). The relative importance of each variable in the generation of climatically suitable zones for the presence of the species under study indicated that the variable Annual Mean Temperature presented a greater contribution for 42.8% of these species. The variables Annual Precipitation, Mean Temperature of Wettest Quarter, and Precipitation of Coldest Quarter presented a higher contribution for the 28.5%, 14.2%, and 7.14% of species under study, respectively. The rest of the variables did not present a marked influence on the generation of climatically suitable zones for the species under study (Table 4).

The models allowed the identification of three groups of species according to the percentage of loss of climatic suitability between current climatic conditions and the three climate change scenarios considered in this work (Fig. S1). In the first group (high vulnerability), the species *C. viridis, C. scutulatus, C. molossus,* and *C. ravus* showed a loss of climatic suitability of between 40% and 66% in at least two climate change scenarios used in this study. In the second group (medium vulnerability), *C. pricei, C. ruber, C. lepidus, C. basiliscus, C. tigris*, and *C. cerastes* showed a loss of climatic suitability of between 1% and 34%. In group three (low vulnerability), the species *C. willardi, C. intermedius, C. enyo,*

**Table 4** **The relative importance values of each variable in the generation of habitat suitability models for the *Crotalus* species under study.** Area under the curve (AUC) values also provided that allow the evaluation of habitat suitability models. Variables are bio1 = Annual Mean Temperature, bio2 = Mean Diurnal Range, bio8 = Mean Temperature of Wettest Quarter, bio12 = Annual Precipitation, bio13 = Precipitation of Wettest Month, bio14 = Precipitation of Driest Month, bio15 = Precipitation Seasonality, bio18 = Precipitation of Warmest Quarter, bio19 = Precipitation of Coldest Quarter.

| Species | bio1 | bio2 | bio8 | bio12 | bio13 | bio14 | bio15 | bio18 | bio19 | AUC |
|---|---|---|---|---|---|---|---|---|---|---|
| *Crotalus atrox* | 8.65 | 1.7 | 4.6 | 45.2 | 12 | 4.4 | 14.8 | 2.7 | 5.6 | 0.8 |
| *C. basiliscus* | 22.2 | 8.8 | 27.4 | 6.4 | 4.7 | 11.4 | 7.3 | 2.2 | 9.4 | 0.8 |
| *C. cerastes* | 39.1 | 5.2 | 6.8 | 4.7 | 10.6 | 10.7 | 5.15 | 13.1 | 4.2 | 0.8 |
| *C. enyo* | 12.4 | 1.25 | 0 | 28.4 | 15.6 | 6.6 | 14.7 | 11 | 9.9 | 0.8 |
| *C. intermedius* | 0 | 13.6 | 0 | 55.4 | 9.3 | 0 | 7.9 | 11.3 | 2.3 | 0.8 |
| *C. lepidus* | 22.7 | 14.5 | 6.9 | 7.8 | 4.7 | 6.5 | 16.3 | 6.6 | 13.8 | 0.8 |
| *C. molossus* | 35.5 | 11.3 | 5.4 | 1.4 | 3.7 | 6.4 | 21.4 | 4.2 | 10.5 | 0.8 |
| *C. pricei* | 39.5 | 0.9 | 3.1 | 0.2 | 3.5 | 7.7 | 12.9 | 11 | 20.5 | 0.9 |
| *C. ravus* | 55.3 | 1 | 7.5 | 13.2 | 2.3 | 10 | 3.1 | 0 | 7.3 | 0.9 |
| *C. ruber* | 11.1 | 0.5 | 29.3 | 6.7 | 3.9 | 17.9 | 11.4 | 14.1 | 4.7 | 0.8 |
| *C. scutulatus* | 3.3 | 12.2 | 36.2 | 6.4 | 7.2 | 7.7 | 12.1 | 7.95 | 6.75 | 0.88 |
| *C. tigris* | 28.7 | 18 | 0.4 | 1.7 | 16.5 | 4.4 | 7.2 | 2.75 | 20.2 | 0.91 |
| *C. viridis* | 49.6 | 16.7 | 2.4 | 4.1 | 1.3 | 4 | 7.7 | 11.45 | 2.55 | 0.95 |
| *C. willardi* | 0 | 1 | 0 | 0 | 31 | 29.35 | 12.8 | 0.05 | 25.65 | 0.93 |

and *C. atrox* showed an increase in climatic suitability for the climate change scenarios considered in this study (Table 5).

# DISCUSSION

*Hutchinson (1957)* defines the niche of a species as a *n*-dimensional space, where each dimension represents the response of a species to the variation of a certain variable. In this way, each site on Earth is characterized by a set of environmental conditions that define a specific habitat inhabited or uninhabited by a community of species (*Kearney, 2006*). In this sense, our results indicate that for current climate conditions, according to the PCA, the climatic profile of the distribution area of the species under study can be viewed from two approaches. The first is approach one (PC1), where the climate profile is determined to a greater extent by the Annual Precipitation. With approach two (PC2), the greatest contribution is provided by the variables Precipitation of Warmest Quarter and Precipitation of Coldest Quarter. For the climate change scenarios used in this study, the variables Annual Precipitation, Precipitation of Warmest Quarter, and Precipitation of Coldest Quarter will continue to make the greatest contribution to the climate profile.

Climate change in the last 30 years has produced numerous changes in the distribution and abundance of species (*Parmesan & Yohe, 2003*; *Root et al., 2003*) and has been implicated in the extinction of several species (*Pounds, Fogden & Campbell, 1999*). For the period 2021–2040, our results of climatic suitability loss identify three levels of vulnerability (high, medium, and low) for the species under study. For the group with high vulnerability, we identified *C. viridis, C. scutulatus, C. molossus*, and *C. ravus*, which represents 28.5%

**Table 5 Three levels of habitat vulnerability for rattlesnakes of the genus *Crotalus* in North America.** Habitat measured in square kilometers (km$^2$), and percentage of change shown to future scenarios. Climate change scenarios correspond to BCC-CSM2-MR, CNRM-CM6-1, IPSL-CM6A-LR.

| Groups | Species | Current weather | BCC-CSM2-MR | CNRM-CM6-1 | IPSL-CM6A-LR |
|---|---|---|---|---|---|
| High vulnerability | *Crotalus viridis* | 1820437 | 639564 | 620547 | 646873 |
| | | Change rate (%) −64.87 | −65.91 | −64.47 | |
| | *C. scutulatus* | 809924 | 382908 | 420207 | 362717 |
| | | Change rate (%) −52.72 | −48.12 | −55.22 | |
| | *C. molossus* | 959356 | 489167 | 458906 | 467002 |
| | | Change rate (%) −49.01 | −52.17 | −51.32 | |
| | *C. ravus* | 44437 | 25707 | 23208 | 25647 |
| | | Change rate (%) −42.15 | −47.77 | −42.28 | |
| Medium vulnerability | *C. pricei* | 146648 | 102440 | 107256 | 96710 |
| | | Change rate (%) −30.15 | −26.86 | −34.05 | |
| | *C. ruber* | 91162 | 65837 | 72058 | 71887 |
| | | Change rate (%) −27.78 | −20.96 | −21.14 | |
| | *C. lepidus* | 577117 | 440588 | 439025 | 443955 |
| | | Change rate (%) −23.66 | −23.93 | −23.07 | |
| | *C. basiliscus* | 78814 | 61888 | 64637 | 63889 |
| | | Change rate (%) −21.48 | −17.99 | −18.94 | |
| | *C. tigris* | 107274 | 93535 | 92460 | 92400 |
| | | Change rate (%) −12.81 | −13.81 | −13.87 | |
| | *C. cerastes* | 262133 | 405465 | 252009 | 258217 |
| | | Change rate (%) 54.68 | −3.86 | −1.49 | |
| Low vulnerability | *C. willardi* | 46803 | 58109 | 67865 | 74058 |
| | | Change rate (%) 24.16 | 45 | 58.23 | |
| | *C. intermedius* | 40922 | 56759 | 57932 | 59345 |
| | | Change rate (%) 38.7 | 41.57 | 45.02 | |
| | *C. enyo* | 42845 | 68720 | 66650 | 63527 |
| | | Change rate (%) 60.39 | 55.56 | 48.27 | |
| | *C. atrox* | 649052 | 1340144 | 1255603 | 1242055 |
| | | Change rate (%) 106.48 | 93.45 | 91.36 | |

of the analyzed species. In the group of medium vulnerability, we identified *C. pricei*, *C. ruber*, *C. lepidus*, *C. basiliscus*, *C. tigris*, and *C. cerastes*, which represent 42.8% of our studied species. The species with low vulnerability includes *C. willardi*, *C. intermedius*, *C. enyo*, and *C. atrox*, representing 28.6% of our considered species. Various authors have pointed out that the breadth of the niche can have an important effect on the risk of extinction of a species because species with broader niches could be less vulnerable to abrupt environmental variation under anthropogenic climate change. At the opposite extreme, species with narrow niches would be particularly threatened by climatic changes (*Brown, 1984*; *Johnson, 1998*; *Kotiaho et al., 2005*; *Pearson et al., 2014*; *Saupe et al., 2015*).

There is substantial evidence from a variety of taxa that supports the theory that narrowed niches drive the risk of extinction of species in the face of climate change variations (*e.g.*,

fishes (*Munday, 2004*), bats (*Boyles & Storm, 2007*), birds (*Seoane & Carrascal, 2008*), frogs (*Botts, Erasmus & Alexander, 2013*), and plants (*Ozinga et al., 2013*). In relation to this, for the period 2021–2040, within the high-vulnerability and medium-vulnerability groups, *C. viridis*, *C. molossus*, *C. tigris*, *C. scutulatus*, *C. ruber*, and *C. cerastes* showed reduced niches for the variables related to temperature. This coincides with the aforementioned predictions since it would be expected that the species under study with reduced niches related to temperature present a greater disturbance in their habitat with respect to the increase in temperature projected for the period 2021–2040. However, other species in these same two groups (*C. ravus*, *C. basiliscus*, and *C. lepidus*) show a greater niche width compared to several species classified in the low-vulnerabilty group (*C. atrox*, *C. pricei*, and *C. intermedius*). This finding contrasts what is proposed above. In this context, *Carrillo-Angeles et al. (2016)* suggest that although various studies reinforce the hypothesis that species with narrow niches are more susceptible to climate change, there is no single trend in the fate of species with narrow niches and their vulnerability to environmental variations. For example, projections for an increase in greenhouse gases and, consequently, in temperature, for the year 2050 in Europe suggest that some of the most affected species will be those that inhabit colder northern regions, species with low densities, and species with less tolerance to aridity (*Huntley et al., 1995*; *Thuiller et al., 2005a*).

Related to this last point, evidence suggests an increase in temperature and low rainfall for the period 2021–2040. For example, the comparison of means indicates that the variables Annual Precipitation, Precipitation of Warmest Quarter, and Precipitation of Coldest Quarter will present a relative stability for the period 2021–2040, with respect to what is shown in the current climate. However, for the variables Annual Mean Temperature, Mean Diurnal Range, Mean Temperature of Wettest Quarter, an increase in the averages of between 1.74 °C and 1.99 °C is expected; 0.11 °C and 0.49 °C, and 1.1 °C and 1.8 °C, respectively. In this regard, various studies have mentioned that the significant increase in temperature and the low availability of water will lead to a reduction in humidity of the air and substrate (*Seager et al., 2007*; *Ye & Grimm, 2013*; *Kunkel et al., 2013*). This is a condition that may have significant detrimental effects on reptiles that are less tolerant to aridity (*Inman et al., 2014*; *Hatten et al., 2016*).

Our results show that for *C. ravus*, *C. basiliscus*, and *C. lepidus*, despite presenting wide climatic niches for the variables related to precipitation and temperature, their ideal habitat is influenced to a greater extent the Annual Mean Temperature and Mean Temperature of Wettest Quarter, respectively. Like the rest of the species classified as high and medium vulnerability, they are also influenced to a greater extent by the variables Annual Mean Temperature and Mean Temperature of Wettest Quarter. In contrast, for *C. atrox*, *C. enyo*, *C. willardi*, and *C. intermedius*, four species identified with low vulnerability to climate change, the variables related to temperature show little contribution to the generation of suitable climatic environments for their distribution. In this way, the evidence suggests that for our species identified with high vulnerability to climate change, they can be considered as less tolerant to the increase in aridity projected for the period 2021–2040.

In conclusion, the increase in the variables Annual Mean Temperature and Mean Temperature of Wettest Quarter may compromise the climatic suitability of at least 71.4%

of the species considered in our study. In this sense, for the species under study, the niche width, by itself, cannot be considered as a determining factor that helps to predict the vulnerability of their climatic suitability under rapid environmental change. However, evidence from our study shows how the relative importance of climatic variables in the construction of niche modeling can help us understand the vulnerability of the climatic suitability of the species under study to global climate change.

In this study, we used correlative methods to model the climatic suitability of the species under study and estimate niche width. *Soberón (2007)* pointed out that the realized niche is determined by biotic restrictions in the fundamental ecophysiological niche, population dynamics (*e.g.,* source–sink dynamics) and dispersion limitations (that is, accessibility). Therefore, in our study, we are not considering the physiological limits of the species and, although *Cuervo-Robayo et al. (2017)* comment that correlative ecological niche models are a good technique to capture exposure to climate change, we cannot rule out that we could be underestimating or overestimating our results. However, mechanistic (physiological) methods can also be subject to overestimation or underestimation of the niche (*Peterson & Holt, 2003*; *Strubbe et al., 2015*).

## ACKNOWLEDGEMENTS

We thank Edmundo Pérez Ramos and Adrian Nieto Montes de Oca for their logistic help reviewing specimens from the Collection of the Museo de Zoología, Facultad de Ciencias (MZFC), and Víctor Hugo Reynoso for allowing us to review the specimens under his care in the Colección Nacional de Anfibios y Reptiles, Instituto de Biología (CNAR-IBH), both from the Universidad Nacional Autónoma de México. We also thank Héctor Rafael Eliosa León for his logistic help reviewing specimens from the Colección Herpetológica, Facultad de Ciencias Biológicas from the Benemérita Universidad Autónoma de Puebla. We thank Margaret Schroeder for revising the language of the manuscript, and Ferdinand Torres Angeles and Jesús Martín Castillo Cerón for logistical support in carrying out the project. We thank Programa para el Desarrollo Profesional Docente (PRODEP) at Universidad Autónoma del Estado de Hidalgo. We are also thankful for the comments and suggestions provided by three anonymous reviewers that helped us greatly improve our manuscript.

### Funding

This study was supported by Comisión Nacional para el Conocimiento y Uso de la Biodiversidad (CONABIO) projects R045 and JM001, and Fondo Mixto-Comisión Nacional de Ciencia y Tecnología (Fomix-CONACyT) 191908 Biodiversidad del Estado de Hidalgo-3a. The funders had no role in study design, data collection and analysis, decision to publish, or preparation of the manuscript.

### Grant Disclosures

The following grant information was disclosed by the authors:

Comisión Nacional para el Conocimiento y Uso de la Biodiversidad (CONABIO) projects R045 and JM001.
Fondo Mixto-Comisión Nacional de Ciencia y Tecnología (Fomix-CONACyT) 191908 Biodiversidad del Estado de Hidalgo-3a.

## Competing Interests

The authors declare there are no competing interests.

## Author Contributions

- Jorge Luis Becerra-López conceived and designed the experiments, performed the experiments, analyzed the data, prepared figures and/or tables, authored or reviewed drafts of the paper, and approved the final draft.
- Raciel Cruz-Elizalde and Aurelio Ramírez-Bautista conceived and designed the experiments, performed the experiments, prepared figures and/or tables, authored or reviewed drafts of the paper, and approved the final draft.
- Itzel Magno-Benítez and Claudia Ballesteros-Barrera conceived and designed the experiments, performed the experiments, authored or reviewed drafts of the paper, and approved the final draft.
- Javier Alvarado-Díaz, Robert W. Bryson Jr, Uriel Hernández-Salinas and Christian Berriozabal-Islas performed the experiments, authored or reviewed drafts of the paper, and approved the final draft.
- César A. Díaz-Marín performed the experiments, prepared figures and/or tables, authored or reviewed drafts of the paper, and approved the final draft.
- Karen Fraire-Galindo, Juan Tello-Ruiz and Alexander Czaja performed the experiments, analyzed the data, prepared figures and/or tables, authored or reviewed drafts of the paper, and approved the final draft.
- María Guadalupe Torres-Delgado analyzed the data, prepared figures and/or tables, authored or reviewed drafts of the paper, and approved the final draft.

## Data Availability

The geographic records of the 14 species of *Crotalus* analyzed in this study are available in the Supplemental File.

## Supplemental Information

Supplemental information for this article can be found online at http://dx.doi.org/10.7717/peerj.13154#supplemental-information.

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
