# Peer review of "Does size matter? An analysis of the niche width and vulnerability to climate change of fourteen species of the genus Crotalus from North America"

_PeerJ, doi:10.7717/peerj.13154_

## Round 0.1 · original submission · Major Revisions

Dear Dr. Becerra-López,

According to the reviewers, the manuscript provides an interesting examination of whether the vulnerability to climate change is related to niche width of several species of Crotalus. However, they find some major concerns.

The first major issue concerns the description and justification of the methods that it is not adequate to ensure replicability. In this regard, I strongly suggest you take into account the article of Araújo and co-authors (Araújo et al Standards for distribution models in biodiversity assessment. Sci. Adv. 2019; 5 : eaat4858) recommending best-practice standards and detailed guidelines to perform niche analysis.

The second major issue concerns the discussion that must be improved to be located in the context of threats due to climate change. So, I encourage you to improve the manuscript according to the reviewers' suggestions. Please, respond point-to-point to the comments of reviewers to speed up the process of revision.

Once again, thank you for submitting your manuscript to PeerJ and we look forward to receiving your revision.

Sincerely,
Gabriele Casazza

Reviewer 1 ·

Basic reporting

Generally, your study is interesting, well thought out, and well written, and has been a pleasure to review. It is a complete body of work worthy of publication. I do have some general suggestions for the written aspects of your study which could strengthen your work if clarified.

Line 91-94: ‘Climatic variations’ is a little bit vague, and you could make this statement for most species – could you please be more specific here, and mention some examples of exactly which climatic variables are important for this genus. The second half of this sentence also is a bit vague – when you say ‘precise effects’, are you talking about how climate change might vary these climatic variables? If so, please specify this, and perhaps give examples of how the specific variables (or at least a subset of these) are predicted to change in study region under climate change (this might mean re-writing the following sentence to flow on from this).

Line 120: GBIF is ‘Global Biodiversity Information Facility’

Throughout the manuscript, after the initial introduction in the methods, it would be of greater clarity to the reader to refer to your variables by their full (or abbreviated) name. For example, rather than ‘bio12’, say ‘annual precipitation’ (or even ‘AP’, after first spelling out annual precipitation).

Line 249: Perhaps you could change this reference to the most recent IPCC report?

Line 260-261: How did you define the three levels of vulnerability?

Line 114: Is the presence data you collated from various sources (excluding GBIF) shared?

In general, your writing is clear, however there are some grammatical errors throughout. Some sections, especially the discussion, will need some restructuring, so I have just given some examples of things to look out for:
Line 80-81: Change to …provides a predictive measure about how the habitat suitability…
Line 101: lower case ‘t’ needed after colon. Change order of sentence to ‘…question arises: can the breadth of niche, by itself…’
Line 151: change ‘was’ to ‘were’
Line 246: I’m not sure the wording is right here. Perhaps: ‘…present a relative stable contribution…’. I’m not 100% clear on what you are saying here.

Perhaps as a supplementary figure, could you provide spatial projections of your SDM for each species to complement Figure 1? This would not be essential to publication, however would be beneficial to your overall body of work.

Experimental design

Your general experimental approach is robust and clearly explained. There are some decisions that you have made throughout that need more justification, and there are some approaches to your design that could be amended to produce more robust results. Below, I have outlined some considerations you could take with aspects of your experiment, and also asked for some clarification on some aspects. This may result in you needing to adjust some of your experimental paramaters and re-run analysis.

Presence data: did you take steps to ensure that spatial projections were identical for all data points, and that there were no duplicate records in your data? See Gueta, T., & Carmel, Y. (2016). Quantifying the value of user-level data cleaning for big data: A case study using mammal distribution models. Ecological Informatics, 34, 139-145 for some examples of steps you may want to take in ensuring your presence data is accurate.

Climatic variables: Your methods for detecting collinearity are robust. However, you have not provided biological reasoning for why you included each of the variables you did. You do mention that Maxent performs more poorly under complex and non-complex models. You have included 9 variables here, which is perhaps more than most studies, and in your results you show some variables to have consistently little input to models – perhaps these could be eliminated, and models re-run. I suggest Elith, J., Phillips, S. J., Hastie, T., Dudík, M., Chee, Y. E., & Yates, C. J. (2011). A statistical explanation of MaxEnt for ecologists. Diversity and distributions, 17(1), 43-57 for further reading about how model selection could be impacted by variable choice.

Also, you only consider the ‘worst case’ ssp585 scenario. Given you are predicting short term, and perhaps this is the most likely outcome given current action on climate change: perhaps this is reasonable, but please provide some clarification and/or justification for this choice, over other scenarios (e.g. ssp370). I presume that you chose this scenario because it would most likely result in the greatest change to species distribution, thus would be the best way to test your hypothesis, but please clarify this.

Niche range: you provide a reference to the method of defining niche range, but could you please add one or two sentences giving a very brief overview of what is involved, given there are many differing ways of calculating niche breadth. Additionally, perhaps you could consider calculating niche hypervolume as a metric of niche breadth as a comparative measure? See Blonder, B., Lamanna, C., Violle, C., & Enquist, B. J. (2014). The n‐dimensional hypervolume. Global Ecology and Biogeography, 23(5), 595-609.

SDM: you do not discuss any steps you took to account for bias in your presence data when sampling background points. Steps should be taken to address this, and the background selected manually rather than using the default maxent settings to account for bias in presence data (see Elith et al, 2011 again, as well as Fourcade, Y., Engler, J. O., Rödder, D., & Secondi, J. (2014). Mapping species distributions with MAXENT using a geographically biased sample of presence data: a performance assessment of methods for correcting sampling bias. PloS one, 9(5), e97122.

You talk about using the ENMeval package. You mention ‘calibration’ and ‘evaluation’ of models. Did this involve using the ENMevaluate function? If so, please mention which feature classes and regularisation multipliers you used. If not, I would suggest that this extra step could be a useful addition to your methodology. Perhaps the results of this process could be added to Table 4?

Regarding your transformation to binary models, did you consistently use one threshold (e.g. maxSSS) or do this differently for each species? Please clarify and provide justification for your choice.

Validity of the findings

Your findings are interesting and contribute to this field of science. However, particularly in your discussion, there are some points you need to address relating to potential caveats with your study to provide greater context for your results.

You should discuss the fundamental caveat that is part of your study – measuring niche breadth, and SDM in general, makes the assumption that a species is occurring at its physiological limits. This means that when you are measuring niche breadth, you are making the assumption that a species would not be able to occur at a level above or below your defined niche. However, if your species isn’t occurring at its limits, due to other, non-climatic reasons such as dispersal barriers, competition and/or under-sampling, for example, you will be definition be underestimating that species’ tolerance of environment. While this doesn’t make your study invalid, you should address this in your discussion when discussing your results. See Bush, A., Mokany, K., Catullo, R., Hoffmann, A., Kellermann, V., Sgro, C., ... & Ferrier, S. (2016). Incorporating evolutionary adaptation in species distribution modelling reduces projected vulnerability to climate change. Ecology letters, 19(12), 1468-1478 for further reading.

Throughout your paper, you often mention ‘habitat’ suitability and change. Please change this throughout to descriptors such as ‘climatic niche’, ‘suitable climatic space’ etc. Habitat encompasses many aspects of the landscape, such as land use, vegetation, etc., but you only consider climatic variables – this is an important distinction to make.

Paragraph 2 of discussion – I am not sure this is contributing any new information, as it is confirming other studies, and also you are not using climate models you have developed, but rather just interpreting public data. This needs to be re-written to focus more on how your results are showing climate change with more explicit reference to your specific species. Also, this is a minor point, but you say ‘throughout the current century’, but you are only considering to 2040 in your study.

Additional comments

Thank you for the chance to review your study. It is an interesting piece of science. I hope that my comments will be able to improve your work.

Reviewer 2 ·

Basic reporting

This paper on the correlation between niche width and susceptibility to habitat loss under climate change for multiple species of rattlesnakes was, overall, very clear (except in a few places as noted below) and well written. The authors have done an excellent job reviewing the literature and have included all the key papers that I am aware of. All the figure and tables were clear.
This piece was a coherent unit – and all relevant results were presented.

I saw the supplemental table that included the collections where specimen localities originated, but I did not see any raw data with exact latitude and longitude for each record. Including that would fulfil the Peer J data sharing policy (I believe).

Experimental design

The design falls within the Aims and Scope of PeerJ, and the research question was clear and compelling and addresses a pressing issue of adaptability of an underappreciated taxonomic group. Overall, most of the methods were appropriate and clear enough to allow replication of the results. I had a few questions, mostly to help with clarity.

Line 106 – was there a particular reason why you chose these particular 14 species?
Line 136 – the sample size for each species was perfectly appropriate for Maxent, but I was wondering whether you could say that these points were well distributed across the known range? That is, do you think the species distribution records accurately reflect the total range of each species, or might some species have an under-sampled range?
Line 147 – could you briefly explain what ssp585 is? You mention it in the legend, but not in the text.

Validity of the findings

Their findings seem valid for the methods used.

Additional comments

There were several very minor typos throughout; I highlighted a few that I noticed below.

Line 80: I believe this should read “a predictive measure about how the habitat suitability of a species”
Line 91: I believe this should read “with the greatest number found in Mexico”
Line 101: Rephrasing the phrase to “Can the breadth…” would help with clarity.
Line 203: Table 2, not Tables 2 as written
Line 245: Could you clarify here that bio 12 and 18 are variables related to precipitation, while 1, 2 and 8 are temperature variables?

Reviewer 3 ·

Basic reporting

This manuscript by Becerra-López and colleagues, analyses the vulnerability to climate change related to niche width of 14 species of Crotalus. The authors performed typical correlative models with Maxent, using occurrence records, bioclimatic layers and three global climate models.

My main concern is the selection of the study model. It is not clear why they selected this genus and in particular those 14 species. The work does not indicate specific reasons. The premise on which this manuscript is based seems coherent and interesting to me, however, it is poorly justified. This is a comparative study between species, however, nowhere in the manuscript do the authors address the phylogenetic context, which can help to understand their results. Therefore, despite his descriptive analysis of the breadth of niches, his results are not robust.

After reading it carefully, this manuscript is written in an extremely simple way and throughout the text, it is not very descriptive. There are no details and justify why this genus, species, methods, algorithm, global climate models, scenarios, etc. Why only one scenario (SSPs-585)? This is a serious disadvantage in terms of replicability. Otherwise, it is not comparable to future work. I found the discussion superficial and uninteresting. At the end of my recommendations, I list some references using snakes as a study model in the context of climate change (not cited here).

Besides I consider the information provided in the present manuscript should be highlighted considering the present state of the reptiles all over the world and the extinction risks due to global warming. This work is not located in this context; it could be exploited in the introduction and discussion sections. While there is not enough information for snakes, in general, there is more information for lizards, which share natural history traits.

Finally, the direction of the new wave of modeling and extinction risk is towards projections that can be verifiable. For example, supported by demographic, physiological data, or even with local present extirpations. Otherwise, these are only projections with high levels of uncertainty. Despite being correlative models, which have less power of predictability (according to many authors), they can be related to possible current extirpations or populations at risk of extinction. On the other hand, in this work the tendencies in the distribution of the species are not detailed in terms of the direction of possible expansions or extirpations according to the suitable habitat, its results are based only on percentages. This makes it impossible for decision-makers to use your work as a reference.

Experimental design

See previous comments

Validity of the findings

See previous comments

Additional comments

I have additional comments that will help to improve this manuscript for future versions.

Introduction
Line 71-72. The authors emphasize “tolerances”, however, I didn't see any discussion of this.

82-84 “most niche models”, correlative models have been widely used, but this is not a solid justification to be used.

Line 88. Authors mention that Crotalus is a good model, but the justification is not clear why. Why those species and not others? Many other groups, even reptiles, are better models, for example lizards. Authors should better justify this part. It seems that authors first selected the group and then established the study question. There is no solid research design. I believe that the genus Crotalus would be a good model if there were demographic or physiological data, or vulnerable population info to validate their results.

Line 90. “There are 53 species” Does this argument make it a good study model?

Line 93. "Precise"? is it the correct word?

Lines 96-98. Here is one of my main concerns. Species selected by the authors. On the one hand, the premise on the effects of climate change and the niche breadth is interesting. However, there is no ecological, evolutionary, physiological or biogeographic justification/context for why these 14 species. It gives the impression that the species selection was a priori or subjective. I was surprised not to see "microendemic" species from islands or others from the Yucatan peninsula. For this reason, the results could be biased. Authors should work on this point.

I am not familiar with the physiology of Crotalus, however its tolerance limits could be informative and would support your ecological niche breadth results. Physiological references are needed to support this research.

Line 104. "precise effects" how do you know that your projections are accurate?

Line 109-111. Although the authors mention: "This information is of great relevance for the establishment and development of conservation strategies for species of the genus Crotalus", in the results or conclusions section there are no figures, models, or particular projections that specify which are the sites of greatest vulnerability in the genus or in each species of study. It is difficult for decision-makers to interpret the results as they are presented. The only figure is not informative.

Line 120. Do you mean “Global Biodiversity Information Facility”?

Line 120. Take a look how to cite GBIF (https://www.gbif.org/citation-guidelines).

123-126. I wonder what would happen if they had integrated "microendemic" species such as Crotalus lanomi, C. stejnegerí, or some other island in the Gulf of Calfornia (C. estebanensis, C. catalinensis).

127-128. Regarding "precise" models. The 2.5º resolution may be adequate in species with a wide distribution, however, for species with restricted distribution, the resolution at 30 sec (~ 1km) would be better. I understand that new layers are not yet available on WClim. Subscalling?

147. It must be justified why they used these GCMs.

147, 168-169 and 191-193. In the M&M section authors used three different global climate models for “Climate Profile and Niche Range” and “Vulnerability of habitat suitability to climate change”: BCC-CSM-MR, CNRM-CM6-1 and IPSL-CM6A-LR. However, different ones are mentioned in the results section: GISS-E2-R and MIROC-ESM. In addition, the methodology should be more carefully and deeply explained, for example Becerra-López et al. (2020) (Line 152).

Lines 168 do you mean: BCC-CSM”2”-MR?

Line 171-172. In this section all the details about the species distribution models should be included. For example, replicates, runs, test and training percentage, cross-validation/bootsrapping/subsampling, considerations for best models, AUC or TSS values, thersholds criteria, etc. This section is poorly detailed. With this information it is impossible to replicate it.

190. “Scenario” or global climate model?

Discussion.
The discussion superficial and uninteresting.

234-244 You need to insert a new first paragraph. The first paragraph of a discussion should be a summary of the most important points the reader should take away from your results. That first paragraph of the discussion sets the stage for the following paragraphs. Subsequent paragraphs should then put those results into context of relationships to previously published literature (is it consistent with that literature or if not, does it indicate a new understanding?).

255 Your results suggest: “low rainfall will continue”? On line 245-246 (results) you say “…the comparison of means suggests that the variables bio12, bio18 and bio19 will present a relative stability for the period 2021–2040”. It seems to be a misinterpretation of the results.

257-259. This is a very general pattern. The authors could focus their discussion on ectotherms or reptiles.

266-269. Here the authors repeat the premise of the introduction, but it is not discussed.

269. They need references to support this claim.

Figure 1.
I am very concerned about this single figure. This figure presented here is unpublishable. It is not necessary to show Alaska and Northern Canada. I recommend zooming in on the maps. The species with the northernmost distribution is C. viridis (M), this could be the limit extension for all the panels. With the zoom, the distribution of the species can be better observed, for example, E, I, L and N. Finally, if the arrangement by the journal allows it, it could be a vertical figure, for example, 3 columns and 5 rows. It is necessary to work on the art of the figure. I also think that figures on the effects of climate change on the distribution of species would be useful. Otherwise their results in percentages are not very useful in conservation strategies.

308-311. Would the results have changed if the authors had selected a wider range of species, eg "microendemic" (C. stejnegeri) or island endemic?

Extra references that could be used.
Aubret, F. & Shine, R. (2010). Thermal plasticity in young snakes: how will climate change affect the thermoregulatory tactics of ectotherms? J. Exp. Biol. 213, 242–248.

Bombi P, Capula M, D’Amen M, Luiselli L (2011) Climate change threatens the survival of highly endangered Sardinian populations of the snake Hemorrhois hippocrepis. Animal Biology, 61, 239–248.

Cabrelli, A. L., Stow, A. J. & Hughes, L. A (2014) framework for assessing the vulnerability of species to climate change: a case study of the Australian elapid snakes. Biodivers. Conserv. 23, 3019–3034.

Capula M, Rugiero L, Capizzi D et al. (2014) Long-term, climate change-related shifts in monthly patterns of roadkilled Mediterranean snakes (Hierophis viridiflavus). Herpetological Journal, 24, 97–102.

Freedman AH, Buermann W, Lebreton M, Chirio L, Smith TB (2009) Modeling the Effects of Anthropogenic Habitat Change on Savanna Snake Invasions into African Rainforest. Conservation Biology, 23, 81–92.

Mesquita, P. C. M. D., Pinheiro-Mesquita, S. F. & Pietkzac, C. Are common species endangered by climate change? Habitat suitability projections for the royal ground snake, Liophis reginae (Serpentes, Dipsadidae). North-Western J. Zool. 9, 51–56 (2013).

Moreno-Rueda G, Pleguezuelos JM, Alaminos E (2009) Climate warming and activity period extension in the Mediterranean snake Malpolon monspessulanus. Climatic Change, 92, 235–242.

Nori J, Carrasco PA, Leynaud GC (2014) Venomous snakes and climate change: ophidism as a dynamic problem. Climatic Change, 122, 67–80.

Pomara LY, Ledee OE, Martin KJ, Zuckerberg B (2014) Demographic consequences of climate change and land cover help explain a history of extirpations and range contraction in a declining snake species. Global Change Biology, 20, 2087–2099.

Rugiero L, Milana G, Petrozzi F, Capula M, Luiselli L (2013) Climate-change-related shifts in annual phenology of a temperate snake during the last 20 years. Acta Oecologica, 51, 42–48.

Sahlean TC, Gherghel I, Papes M, Strugariu A, Zamfirescu SR (2014) Refining Climate Change Projections for Organisms with Low Dispersal Abilities: A Case Study of the Caspian Whip Snake. PLOS ONE, 9.

Winter, M., Fiedler, W., Hochachka, W. M., Koehncke, A., Meiri, S. & De la Riva, I. (2016). Patterns and biases in climate change research on amphibians and reptiles: a systematic review. Royal Society Open Science, 3(9), 160158.

I found it surprising that the authors did not discuss their results with this paper (or even other viviparous species):

Lourenço-de-Moraes, R., Lansac-Toha, F. M., Schwind, L. T. F., Arrieira, R. L., Rosa, R. R., Terribile, L. C., Lemes, P., Fernando Rangel, T., Diniz-Filho, J. A. F., Bastos, R. P. & Bailly, D. (2019). Climate change will decrease the range size of snake species under negligible protection in the Brazilian Atlantic Forest hotspot. Scientific Reports, 9(1).

---

## Round 0.2 · Minor Revisions

Dear Dr. Ramírez-Bautista,

The reviewers found the manuscript strongly improved. One of them asks for a few changes to correct typos. Please correct the typos and check for other possible typos, before the acceptance.

Once again, thank you for submitting your manuscript to PeerJ and we look forward to receiving your revision.

Sincerely,
Gabriele Casazza

Reviewer 1 ·

Basic reporting

The authors have addressed each of my points from the first revision. They have also addressed points from the other reviewers to a great standard, and I have no major comments. Some very minor typographical issues I noticed during my review:

Line 128: Typo (‘futher’ -> ‘further’)
Line 136: change ‘recorded’ to ‘obtained’ or similar.
Line 139-140: Perhaps change ‘We carried out a reduction in the number of variables under the criterion that the most robust sets of variables were those that had a direct interaction with the species. These variables were chosen on the basis of ecological theory, and subsequently reduced, when necessary, by statistical analysis (Austin 2007).’ To
We selected a subset of these variables on the basis of ecological theory, and subsequently reduced, when necessary, by statistical analysis (Austin 2007).
Line 161: move your reference for R to the first mention in the paper (Line 127). Also, you have referenced different versions of R – double check which version is correct, and I believe you only need to mention the version in the first instance (for all subsequent times you mention R, simply write ‘R’)
Line 203-205: there is some repetition here from earlier in the methods, you can simplify as you have already said which models you used and where they came from.
Line 322: add a reference for this claim.

Experimental design

no comment

Validity of the findings

no comment

Additional comments

The authors have greatly improved their paper in the review process, and have addressed the suggestions of each of the reviewers to a great standard.

Reviewer 2 ·

Basic reporting

I appreciated having the change to re-review this interesting work. Overall, I thought that the revisions were a big improvement in the clarity and utility of the paper. I only have a few minor suggestions here:
1) Line 129: You say that you assessed sampling bias, but I wasn’t sure what that meant. Did you address observed sampling bias in some way?
2) Line 168: I believe tukey’s post hoc should be capitalized here
3) Line 184: You say that you choose this climate scenario to test your hypothesis “in a better way”. I see what you are saying here, and that this comes from a reviewer’s suggestion, but I think you can phrase it slightly better. Something like “using the most extreme climate scenario provides the best test of our hypothesis as this will result in the largest difference between current and future environmental conditions” or something like that.

Experimental design

No Comment

Validity of the findings

No Comment

Additional comments

I am satisfied with how the authors have addressed my original comments.

---

## Round 0.3 · accepted · Accept

Dear Dr. Ramírez-Bautista,

I am very pleased to inform you that your paper "Does size matter? An analysis of the niche width and vulnerability to climate change of fourteen species of the genus Crotalus from North America" is accepted for publication in the PeerJ. Congratulations!

Thank you for submitting your work to PeerJ.

Sincerely,
Gabriele Casazza